# The Variational Bandwidth Bottleneck: Stochastic Evaluation on an Information Budget

**Anirudh Goyal[1], Yoshua Bengio[1], Matthew Botvinick[2], Sergey Levine[3]**

## Abstract

In many applications, it is desirable to extract only the relevant information from complex input data, which involves making a decision about which input features are relevant. The information bottleneck method formalizes this as an information-theoretic optimization problem by maintaining an optimal tradeoff between compression (throwing away irrelevant input information), and predicting the target. In many problem settings, including the reinforcement learning problems we consider in this work, we might prefer to compress only part of the input. This is typically the case when we have a standard conditioning input, such as a state observation, and a "privileged" input, which might correspond to the goal of a task, the output of a costly planning algorithm, or communication with another agent. In such cases, we might prefer to compress the privileged input, either to achieve better generalization (e.g., with respect to goals) or to minimize access to costly information (e.g., in the case of communication). Practical implementations of the information bottleneck based on variational inference require access to the privileged input in order to compute the bottleneck variable, so although they perform compression, this compression operation itself needs unrestricted, lossless access. In this work, we propose the variational bandwidth bottleneck, which decides for each example on the estimated value of the privileged information before seeing it, i.e., only based on the standard input, and then accordingly chooses stochastically, whether to access the privileged input or not. We formulate a tractable approximation to this framework and demonstrate in a series of reinforcement learning experiments that it can improve generalization and reduce access to computationally costly information.

## 1 Introduction

A model that generalizes effectively should be able to pick up on relevant cues in the input while ignoring irrelevant distractors. For example, if one want to cross the street, one should only pay attention to the positions and velocities of the cars, disregarding their color. The information bottleneck (Tishby et al., 2000) formalizes this in terms of minimizing the mutual information between the *bottleneck* representation layer with the input, while maximizing its mutual information with the correct output. This type of input compression can improve generalization (Tishby et al., 2000), and has recently been extended to deep parametric models, such as neural networks where it has been shown to improve generalization (Achille & Soatto, 2016; Alemi et al., 2016).

The information bottleneck is generally intractable, but can be approximated using variational inference (Alemi et al., 2016). This variational approach parameterizes the information bottleneck model using a neural network (i.e., an encoder). While the variational bound makes it feasible to train (approximate) information bottleneck layers with deep neural networks, the encoder in these networks – the layer that predicts the bottleneck variable distribution conditioned on the input – must still process the full input, before it is compressed and irrelevant information is removed. The encoder itself can therefore fail to generalize, and although the information bottleneck minimizes mutual information with the input on the *training data*, it might not compress successfully on new inputs. To

---

[1] Mila, University of Montreal,[2] Deepmind, [3] University of California, Berkeley, :anirudhgoyal9119@gmail.com

address this issue, we propose to divide our input into two categories: *standard input* and *privileged input*, and then we aim to design a bottleneck that does not need to access the privileged input before deciding how much information about the input is necessary. The intuition behind not accessing the privileged input is twofold: (a) we might want to avoid accessing the privileged input because we want to generalize with respect to it (and therefore compress it) (b) we actually would prefer not to access it (as this input could be costly to obtain).

The objective is to minimize the conditional mutual information between the bottleneck layer and the privileged input, given the standard input. This problem statement is more narrow than the standard information bottleneck, but encompasses many practical use cases. For example, in reinforcement learning, which is the primary subject of our experiments, the agent can be augmented with some privileged information in the form of a model based planner, or information which is the result of communication with another agent. This "additional" information can be seen as a privileged input because it requires the agent to do something extra to obtain it.

Our work provides the following contributions. First, we propose a *variational bandwidth bottleneck (VBB)* that does not look at the privileged input before deciding whether to use it or not. At a high level, the network is trained first to examine the standard input, and then stochastically decide whether to access the privileged input or not. Second, we illustrate several applications of this approach to reinforcement learning, in order to construct agents that can stochastically determine when to evaluate costly model based computations, when to communicate with another agent, and when to access the memory. We experimentally show that the proposed model produces better generalization, as it learns when to use (or not use) the privileged input. For example, in the case of maze navigation, the agent learns to access information about the goal location only near natural bottlenecks, such as doorways.

## 2 PROBLEM FORMULATION

We aim to address the generalization issue described in the introduction for an important special case of the variational information bottleneck, which we refer to as the conditional bottleneck. The conditional bottleneck has two inputs, a standard input, and a privileged input, that are represented by random variables $\mathbf{S}$ and $\mathbf{G}$, respectively. Hence, $\mathbf{S}, \mathbf{G}, \mathbf{Y}$ are three random variables with unknown distribution $\mathbf{p_{dist}}(\mathbf{S}, \mathbf{G}, \mathbf{Y})$.

The information bottleneck provides us with a mechanism to determine the correct output while accessing the minimal possible amount of information about the privileged input $\mathbf{G}$. In particular, we formulate a *conditional variant* of the information bottleneck to minimize the mutual information between the bottleneck layer and the privileged input $\mathbf{I}(\mathbf{Z}, \mathbf{G}|\mathbf{S})$, given the standard input while avoiding unnecessary access to privileged input $\mathbf{G}$. The proposed model consists of two networks (see Fig. 1): The encoder network that takes in the privileged input $\mathbf{G}$ as well as the standard input $\mathbf{S}$ and outputs a distribution over the latent variable $\mathbf{z}$ such that $\mathbf{z} \sim \mathbf{p}(\mathbf{Z}|\mathbf{G}, \mathbf{S})$. The decoder network $\mathbf{p_{dec}}(\mathbf{Y}|\mathbf{Z}, \mathbf{S})$ takes the standard input $\mathbf{S}$ and the compressed representation $\mathbf{Z}$ and outputs the distribution over the target variable $\mathbf{Y}$.

## 3 VARIATIONAL BOTTLENECK ON STANDARD INPUT AND PRIVILEGED INPUT

The information bottleneck (IB) objective (Tishby et al., 2000) is formulated as the maximization of $\mathbf{I}(\mathbf{Z}; \mathbf{Y}) - \beta\mathbf{I}(\mathbf{Z}; \mathbf{X})$, where $\mathbf{X}$ refers to the input signal, $\mathbf{Y}$ refers to the target signal, $\mathbf{Z}$ refers to the compressed representation of $\mathbf{X}$, and $\beta$ controls the trade-off between compression and prediction. The IB has its roots in channel coding, where a compression metric $\mathbf{I}(\mathbf{Z}; \mathbf{X})$ represents the capacity of the *communication channel* between $\mathbf{Z}$ and $\mathbf{X}$. Assuming a prior distribution $\mathbf{r}(\mathbf{Z})$ over the random variable $\mathbf{Z}$, constraining the channel capacity corresponds to limiting the information by which the posterior $\mathbf{p}(\mathbf{Z}|\mathbf{X})$ is permitted to differ from the prior $\mathbf{r}(\mathbf{Z})$. This difference can be measured using the Kullback-Leibler (KL) divergence, such that $\mathbf{D}_{\mathrm{KL}}(\mathbf{p}(\mathbf{Z}|\mathbf{X})\|\mathbf{r}(\mathbf{Z}))$ refers to the channel capacity.

Now, we write the equations for the variational information bottleneck, where the bottleneck is learnt on both the standard input $\mathbf{S}$ as well as a privileged input $\mathbf{G}$. The Data Processing Inequality (DPI) (Cover & Thomas, 2006) for a Markov chain $\mathbf{x} \to \mathbf{z} \to \mathbf{y}$ ensures that $\mathbf{I}(\mathbf{x}; \mathbf{z}) \geq \mathbf{I}(\mathbf{x}; \mathbf{y})$. Hence for a bottleneck where the input is comprised of both the standard input as well as privileged input, we have $\mathbf{I}(\mathbf{Z}; \mathbf{G}|\mathbf{S}) \geq \mathbf{I}(\mathbf{Y}; \mathbf{G}|\mathbf{S})$. To obtain an upper bound on $\mathbf{I}(\mathbf{Z}; \mathbf{G}|\mathbf{S})$, we must first obtain an

upper bound on $\mathbf{I}(\mathbf{Z}; \mathbf{G}|\mathbf{S} = \mathbf{s})$, and then average over $\mathbf{p}(\mathbf{s})$. We get the following result: We ask the reader to refer to the section on the conditional bottleneck in the supplementary material for the full derivation.

$$\mathbf{I}(\mathbf{Z}; \mathbf{G}|\mathbf{S}) \leq \sum_s \mathbf{p}(\mathbf{s}) \sum_g \mathbf{p}(\mathbf{g}) \mathbf{D}_{\mathrm{KL}}(\mathbf{p}(\mathbf{Z}|\mathbf{s}, \mathbf{g}) \| \mathbf{r}(\mathbf{Z})) \tag{1}$$

# 4 THE VARIATIONAL BANDWIDTH BOTTLENECK

We now introduce our proposed method, the variational bandwidth bottleneck (VBB). The goal of the variational bandwidth bottleneck is to avoid accessing the privileged input $\mathbf{G}$ if it is not required to make an informed decision about the output $\mathbf{Y}$. This means that the decision about whether or not to access $\mathbf{G}$ must be made only on the basis of the standard input $\mathbf{S}$. The standard input is used to determine a channel capacity, $\mathbf{d_{cap}}$, which controls how much information about $\mathbf{G}$ is available to compute $\mathbf{Z}$.

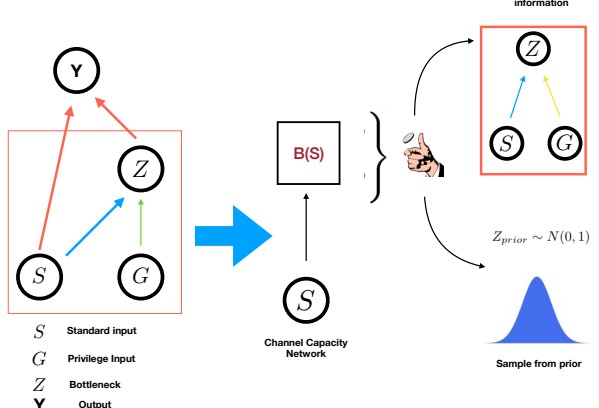

Figure 1: The variational bandwidth bottleneck: Based on the standard input $S$, the channel capacity network determines the capacity of the bottleneck $Z$. The channel capacity then determines the probability of accessing the privileged input. In the event that the privileged input is not accessed, no part of the model actually reads its value.

If $\mathbf{d_{cap}}$ denotes the channel capacity, one way to satisfy this channel capacity is to access the input losslessly with probability $\mathbf{d_{cap}}$, and otherwise send no information about the input at all. In this communication strategy, we have $\mathbf{p}(\mathbf{Z}|\mathbf{S}, \mathbf{G}) = \delta(\mathbf{f_{enc}}(\mathbf{S}, \mathbf{G}))$ if we choose to access the privileged input (with probability $\mathbf{d_{cap}}$), where $\mathbf{f_{enc}}(\mathbf{S}, \mathbf{G})$ is a deterministic encoder, and $\delta$ denotes the Dirac delta function. The full posterior distribution $\mathbf{p}(\mathbf{Z}|\mathbf{S}, \mathbf{G})$ over the compressed representation can be written as a weighted mixture of (a) (deterministically) accessing the privileged input and standard input and (b) sampling from the prior (when channel capacity is low), such that $\mathbf{z}$ is sampled using

$$\mathbf{z} \sim d_{cap} * (\delta(\mathbf{f_{enc}}(\mathbf{S}, \mathbf{G}))) + (1 - d_{cap}) * \mathbf{r}(\mathbf{z}). \tag{2}$$

This modified distribution $\mathbf{p}(\mathbf{Z}|\mathbf{S}, \mathbf{G})$ allows us to dynamically adjusts how much information about $\mathbf{G}$ is transmitted through $\mathbf{Z}$. As shown in the Figure 1, if $\mathbf{d}_{\mathrm{cap}}$ is set to zero, $\mathbf{Z}$ is simply sampled from the prior and contains no information about $\mathbf{G}$. If it is set to one, the privileged information in $\mathbf{G}$ is deterministically transmitted. The amount of information about $\mathbf{G}$ that is transmitted is therefore determined by $\mathbf{d}_{\mathrm{cap}}$, which will depend only on the standard input $\mathbf{S}$.

This means that the model must decide how much information about the privileged input is required before accessing it. Optimizing the information bottleneck objective with this type of bottleneck requires computing gradients through the term $\mathbf{D}_{\mathrm{KL}}(\mathbf{p}(\mathbf{Z}|\mathbf{S}, \mathbf{G}) \| \mathbf{r}(\mathbf{Z}))$ (as in Eq. 1), where $\mathbf{z} \sim \mathbf{p}(\mathbf{Z}|\mathbf{S}, \mathbf{G})$ is sampled as in Eq. 2. The non-differentiable binary event, whose probability is represented by $\mathbf{d}_{\mathrm{cap}}$, precludes us from differentiating through the channel capacity directly. In the next sections, we will first show that this mixture can be used within a variational approximation to the information bottleneck, and then describe a practical approximation that allows us to train the model with standard backpropagation.

## 4.1 TRACTABLE EVALUATION OF CHANNEL CAPACITY

In this section, we show how we can evaluate the channel capacity in a tractable way. We learn a deterministic function $\mathbf{B}(\mathbf{S})$ of the standard input $\mathbf{S}$ which determines channel capacity. This

function outputs a scalar value for $\mathbf{d_{cap}} \in (\mathbf{0}, \mathbf{1})$, which is treated as the probability of accessing the information about the privileged input. This deterministic function $\mathbf{B(S)}$ is parameterized as a neural network. We then access the privileged input with probability $\mathbf{d_{cap}} = \mathbf{B(S)}$. Hence, the resulting distribution over $\mathbf{Z}$ is a weighted mixture of accessing the privileged input $\mathbf{f_{enc}(S, G)}$ with probability $\mathbf{d_{cap}}$ and sampling from the prior with probability $\mathbf{1 - d_{cap}}$. At inference time, using $\mathbf{d_{cap}}$, we sample from the Bernoulli distribution $\mathbf{b} \sim \mathbf{Bernoulli(d_{prob})}$ to decide whether to access the privileged input or not.

## 4.2 OPTIMIZATION OF THE KL OBJECTIVE

Here, we derive the KL divergence objective that allows for tractable optimization of $\mathbf{D_{KL}(p(Z|S, G)\|r(Z))}$ (as in Eq. 1, 2).

**Proposition 1** *Given the standard input $s$, privileged input $g$, bottleneck variable $z$, and a deterministic encoder $f_{enc}(s, g)$, we can express the $D_{\mathrm{KL}}$ between the weighed mixture and the prior as*

$$D_{\mathrm{KL}}(p(z|s,g)\|r(z)) = -d_{cap}\log d_{cap} + (1 - d_{cap})[\log p(f(s,g)) - \log(d_{cap} * p(f(g,s)) + (1 - d_{cap}))] \tag{3}$$

The proof is given in the Appendix, section B. This equation is fully differentiable with respect to the parameters of $\mathbf{f(g, s)}$ and $\mathbf{B(s)} = \mathbf{d_{cap}}$, making it feasible to use standard gradient-based optimizers.

**Summary:** As in Eq. 2, we approximate $\mathbf{p(Z|S, G)}$ as a weighted mixture of $\mathbf{f_{enc}(S, G)}$ and the normal prior, such that $\mathbf{z} \sim \mathbf{d_{cap}} * (\mathbf{f_{enc}(S, G)}) + (\mathbf{1 - d_{cap}}) * \mathcal{N}(\mathbf{0}, \mathbf{1})$. Hence, the quantity $\mathbf{D_{KL}(p(Z|S, G)\|r(Z))}$ can be seen as a bound on the information bottleneck objective. When we access the privileged input $\mathbf{G}$, we pay a *cost* equal to $\mathbf{I(Z, G|S)}$, which is bounded by $\mathbf{D_{KL}(p(Z|S, G)\|r(Z))}$ as in Eq. 1. Hence, optimizing this objective causes the model to avoid accessing the privileged input when it is not necessary.

## 5 VARIATIONAL BANDWIDTH BOTTLENECK WITH RL

In order to show how the proposed model can be implemented, we consider a sequential decision making setting, though our variational bandwidth bottleneck could also be applied to other learning problems. In reinforcement learning, the problem of sequential decision making is cast within the framework of MDPs (Sutton et al., 1998). Our proposed method depends on two sources of input, standard input and privileged input. In reinforcement learning, privileged inputs could be the result of performing any upstream computation, such as running model based planning. It can also be the information from the environment, such as the goal or the result of active perception. In all these settings, the agent must decide whether to access the privileged input or not. If the agent decides to access the privileged input, then the the agent pays an *"information cost"*. The objective is to maximize the expected reward and reduce the cost associated with accessing privileged input, such that across all states on average, the information cost of using the privileged information is minimal.

We parameterize the agent's policy $\pi_\theta(\mathbf{A|S, G})$ using an encoder $\mathbf{p_{enc}(Z|S, G)}$ and a decoder $\mathbf{p_{dec}(A|S, Z)}$, parameterized as neural networks. Here, the channel capacity network $\mathbf{B(S)}$ would take in the standard input that would be used to determine channel capacity, depending on which we decide to access the privileged input as in Section 4.1, such that we would output the distribution over the actions. That is, $\mathbf{Y}$ is $\mathbf{A}$, and $\pi_\theta(\mathbf{A \mid S, G}) = \sum_{\mathbf{z}} \mathbf{p_{priv}(z \mid S, G)} \mathbf{p_{dec}(A \mid S, z)}$. This would correspond to minimizing $\mathbf{I(A; G|S)}$, resulting in the objective

$$J(\theta) \equiv \mathbb{E}_{\pi_\theta}[\mathbf{r}] - \beta\mathbf{I(A; G \mid S)} \quad = \mathbb{E}_{\pi_\theta}[r] - \beta\mathbf{I(Z; G \mid S)}, \tag{4}$$

where $\mathbb{E}_{\pi_\theta}$ denotes an expectation over trajectories generated by the agent's policy. We can minimize this objective with standard optimization methods, such as stochastic gradient descent with backpropagation.

## 6 Related Work

A number of prior works have studied information-theoretic regularization in RL. For instance, van Dijk & Polani (2011) use information theoretic measures to define relevant goal-information, which then could be used to find subgoals. Our work is related in that our proposed method could be used to find relevant goal information, but without accessing the goal first. Information theoretic measures have also been used for exploration (Still & Precup, 2012; Mohamed & Rezende, 2015; Houthooft et al., 2016; Gregor et al., 2016). More recently Goyal et al. (2019) proposed InfoBot, where "decision" states are identified by training a goal conditioned policy with an information bottleneck. In InfoBot, the goal conditioned policy always accesses the goal information, while the proposed method *conditionally* access the goal information. The VBB is also related to work on conditional computation. Conditional computation aims to reduce computation costs by activating only a part of the entire network for each example (Bengio et al., 2013). Our work is related in the sense that we activate the entire network, but only *conditionally* access the privileged input.

Another point of comparison for our work is the research on attention models ((Bahdanau et al., 2014; Mnih et al., 2014; Xu et al., 2015)). These models typically learn a policy, that allows them to selectively attend to parts of their input. However, these models still need to access the entire input in order to decide where to attend. Our method dynamically decides whether to access privileged information or not. As shown in our experiments, our method performs better than the attention method of Mnih et al. (2014).

Recently, many models have been shown to be effective at learning communication in multi-agent reinforcement learning (Foerster et al., 2016; Sukhbaatar et al., 2016). (Sukhbaatar et al., 2016) learns a deep neural network that maps inputs of all the agents to their respective actions. In this particular architecture, each agent sends its state as the communication message to other agents. Thus, when each agent takes a decision, it takes information from all the other agents. In our proposed method, each agent communicates with other agents only when its necessary.

Our work is also related to work in behavioural research that deals with two modes of decision making (Dic, 1985; Kahneman, 2003; Sloman, 1996; Botvinick & Braver, 2015; Shenhav et al., 2017): an automatic systems that relies on habits and a controlled system that uses some extra information for making decision making. These systems embody different accuracy and demand trade-offs. The habit based system (or the default system) has low computation cost but is often more accurate, whereas the controlled system (which uses some external information) achieves greater accuracy but often is more costly. The proposed model also has two parts of input processing, when the channel capacity is low, agent uses its standard input only, and when channel capacity is high, agent uses both the standard as well as privileged input. The habit based system is analogous to using only the standard input, while the controlled system could be analogous to accessing more costly privileged input.

## 7 Experimental Evaluation

In this section, we evaluate our proposed method and study the following questions:

**Better generalization?**   Does the proposed method learn an effective bottleneck that generalizes better on test distributions, as compared to the standard conditional variational information bottleneck?

**Learn when to access privileged input?**   Does the proposed method learn when to access the privileged input dynamically, minimizing unnecessary access?

### 7.1 Baselines

We compare the proposed method to the following methods and baselines:

**Conditional Variational Information Bottleneck (VIB):**   The agent always access the privileged input, with a VIB using both the standard and the privileged input InfoBot (Goyal et al., 2019).

**Deterministically Accessing Privileged Input (UVFA):**   The agent can deterministically access both the state as well as the privileged input. This has been shown to improve generalization in RL problems UVFA (Schaul et al., 2015).

**Accessing Information at a Cost (AIC):** We compare the proposed method to simpler reinforcement-learning baselines, where accessing privileged information can be formalized as one of the available actions. This action reveals the privileged information, at the cost of a small negative reward. This baseline evaluates whether the explicit VBB formulation provides a benefit over a more conventional approach, where the MDP itself is reformulated to account for the cost of information.

**Randomly accessing goal (RAG)** - Here, we compared the proposed method to the scenario where we randomly access the privileged input (e.g., $50\%$ of the time). This baseline evaluates whether the VBB is selecting when to access the goal in an intentional and intelligent way.

## 7.2 DECIDING WHEN TO RUN AN EXPENSIVE MODEL BASED PLANNER

Model-based planning can be computationally expensive, but beneficial in temporally extended decision making domains. In this setting, we evaluate whether the VBB can dynamically choose to invoke the planner as infrequently as possible, while still attaining good performance. While it is easy to plan using a planner (like a model based planner, which learns the dynamics model of the environment), it is not very cheap, as it involves running a planner at every step (which is expensive). So, here we try to answer whether the agent can decide based on the standard input when to access privileged input (the output of model based planner by running the planner).

**Experimental Setup:** We consider a maze world as shown in Figure 2(a). The agent is represented by a blue dot, and the agent has to reach the goal (represented by a green dot). The agent has access to a dynamics model of the environment (which is pretrained and represented using a parameterized neural network). In this task, the agent only gets a partial view of the surrounding i.e. the agent observes a small number of squares in front of it. The agent has to reach the goal position from the start position, and agent can use the

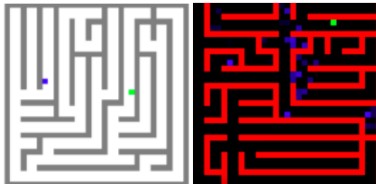

(a) Maze World

Figure 2: Figure on the left shows the environment, where the agent needs to go from blue dot to green dot.

pretrained dynamics model to sample multiple plausible trajectories, and the output of the dynamics model is fed as a conditional input to the agent's policy (similar to (Racanière et al., 2017)), thus the agent can use this dynamics model to predict possible futures, and then make an informed decision based on its current state as well as the result of the prediction from the dynamic model.

In this setup, the current state of the agent (i.e. the egocentric visual observation) acts as the standard input $S$, and the result of running the planner acts as the privileged input $G$. In order to avoid running the model based planner, the agent needs to decide when to access the more costly planner.

| Expensive Inference algorithm | % of times |
|---|---|
| Near the junction | $72\% \pm 5\%$ |
| In the Hallway | $28\% \pm 4\%$ |

Table 1: Running Expensive Model Based Planner

**Results:** - Here, we analyze when the agent access the output of the planner. We find that most of the times agent access the privileged information (output of model based planner) near the junctions as shown in Table 1.

## 7.3 BETTER GENERALIZATION IN GOAL DRIVEN NAVIGATION

The goal of this experiment is to show that, by selectively choosing when to access the privileged input, the agent can generalize better with respect to this input. We consider an agent navigating through a maze comprising sequences of rooms separated by doors, as shown in Figure 7. We use a partially observed formulation of the task, where the agent only observes a small number of squares ahead of it. These tasks are difficult to solve with standard RL algorithms, not only due to the partial observability of the environment but also the sparsity of the reward, since the agent receives a reward only upon reaching the goal (Chevalier-Boisvert et al., 2018). The low probability of reaching the goal randomly further exacerbates these issues. The privileged input in this case corresponds to the

agent's relative distance to the goal $G$. At junctions, the agent needs to know where the goal is so that it can make the right turn. While in a particular room, the agent doesn't need much information about the goal. Hence, the agent needs to learn to access goal information when it is near a door, where it is most valuable. The current visual inputs act as a standard input $S$, which is used to compute channel capacity $d_{cap}$.

**RoomNXSY**

| Train | RoomN6S6 | RoomN12S10 |
|---|---|---|
| RoomN2S4 (UVFA)) | 66% $\pm$ 3% | 49% $\pm$ 3% |
| RoomN2S4 (InfoBot) | 72% $\pm$ 2% | 55% $\pm$ 3% |
| RoomN2S4 (RAG) | 60% $\pm$ 5% | 41% $\pm$ 3% |
| RoomN2S4 (AIC) | 57% $\pm$ 10% | 43% $\pm$ 5% |
| RoomN2S4 (VBB) | **82%** $\pm$ 4% | **60%** $\pm$ 2% |

(a)

**FindObjSY**

| Train | FindObjS7 | FindObjS10 |
|---|---|---|
| FindObjS5 (UVFA)) | 40% $\pm$ 2% | 24% $\pm$ 3% |
| FindObjS5 (InfoBot) | 46% $\pm$ 4% | 22% $\pm$ 3% |
| FindObjS5 (RAG) | 38% $\pm$ 3% | 12% $\pm$ 4% |
| FindObjS5 (AIC) | 39% $\pm$ 2% | 16% $\pm$ 4% |
| FindObjS5 (VBB) | **64%** $\pm$ 3% | **52%** $\pm$ 2% |

(b)

Table 2: Generalization of the agent to larger grids in RoomNXSY envs and FindObj envs. Success of an agent is measured by the fraction of episodes where the agent was able to navigate to the goal in 500 steps. Results are averaged over 500 examples, and 5 different random seeds.

**Experimental setup:** To investigate if agents can generalize by selectively deciding when to access the goal information, we compare our method to InfoBot ((Goyal et al., 2019)) (a conditional variant of VIB). We use different mazes for training, validation, and testing. We evaluate generalization to an unseen distribution of tasks (i.e., more rooms than were seen during training). We experiment on both RoomNXSY ($X$ number of rooms with atmost size $Y$, for more details, refer to the Appendix G) as well as the FindObjSY environment. For RoomNXSY, we trained on RoomN2S4 (2 rooms of at most size

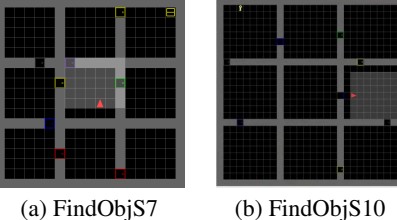

(a) FindObjS7        (b) FindObjS10

Figure 3: Partially Observable FindObjSX environments — The agent is placed in the central room. An object is placed in one of the rooms and the agent must navigate to the object in a randomly chosen outer room to complete the mission. The agent again receives an egocentric observation (7 x 7 pixels), and the difficulty of the task increases with $X$. For more details refer to supplementary material.

6), and evaluate on RoomN6S6 (6 rooms of at most size 6) and RoomN12S10 (12 rooms, of at most size 10). We also evaluate on the FindObjSY environment, which consists of 9 connected rooms of size $Y - 2 \times Y - 2$ arranged in a grid. For FindObjSY, we train on FindObjS5, and evaluate on FindObjS7 and FindObjS10.

**Results:** Tables 3a, 3b compares an agent trained with the proposed method to a goal conditioned baseline (UVFA) (Schaul et al., 2015), a conditional variant of the VIB (Goyal et al., 2019), as well as to the baseline where accessing goal information is formulated as one of the actions (AIC). We also investigate how many times the agent accesses the goal information. We first train the agent on MultiRoomN2S4, and then evaluate this policy on MultiRoomN12S10. We sample 500 trajectories in MultiRoomN12S10env. Ideally, if the agent has learned when to access goal information (i.e., near the doorways), the agent should

| Method | Percentage of times |
|---|---|
| **VBB** | 76% $\pm$ 6% |
| InfoBot (Goyal et al., 2019) | 60% $\pm$ 3% |
| AIC | 62% $\pm$ 6% |

Table 3: **Goal Driven Navigation** - Percentage of time steps on which each method acsess the goal information when the agent is near the junction point (or branching points in the maze. We show that the proposed method learns to access the privileged input (in this case, the goal) only when necessary.

only access the goal information when it is near a door. We take sample rollouts from the pretrained policy in this new environment and check if the agent is near the junction point (or doorway) when the agent access the goal information. Table 3 quantitatively compares the proposed method with

different baselines, showing that the proposed method indeed learns to generalize with respect to the privileged input (i.e., the goal).

## 7.4 LEARNING WHEN TO COMMUNICATE FOR MULTIAGENT COMMUNICATION

Next, we consider multiagent communication, where in order to solve a task, agents must communicate with other agents. Here we show that selectively deciding when to communicate with another agent can result in better learning.

**Experimental setup:** We use the setup proposed by Mordatch & Abbeel (2017). The environment consists of N agents and M landmarks. Both the agents and landmarks exhibit different characteristics such as different color and shape type. Different agents can act to move in the environment. They can also be affected by the interactions with other agents. Asides from taking physical actions, agents communicate with other agents using verbal communication symbols. Each agent has a private goal that is not observed by another agent, and the goal of the agent is grounded in the real physical environment, which might include moving to a particular location. It could also involve other agents (like requiring a particular agent to move somewhere) and hence communication between agents is required. We consider the cooperative setting, in which the problem is to find a policy that maximizes expected return for all the agents. In this scenario, the current state of the agent is the standard input $S$, and the information which might be obtained as a result of communication with other agents is the privileged input $G$. For more details refer to the Appendix (D).

| Model | 6 Agents | 10 agents |
|---|---|---|
| Emergent Communication (Mordatch & Abbeel, 2017) | 4.85 (100%) $\pm$ 0.1% | 5.44 (100%) $\pm$ 0.2% |
| Randomly Accessing (RAG) | 4.95 (50%) $\pm$ 0.2% | 5.65 (50%) $\pm$ 0.1% |
| InfoBot (Goyal et al., 2019) | 4.81 (100%) $\pm$ 0.2% | 5.32 (100%) $\pm$ 0.1% |
| VBB (ours) | **4.72** (23%) $\pm$ 0.1% | **5.22** (34%) $\pm$ 0.05% |

Table 4: **Multiagent communication**: The VBB performs better, as compared to the baselines. In the baseline scenario, all of the agents communicate with all the other agents all the time. Averaged over 5 random seeds.

**Tasks:** Here we consider two tasks: (a) 6 agents and 6 landmarks, (b) 10 agents and 10 landmarks. The goal is for the agents to coordinate with each other and reach their respective landmarks. We measure two metrics: (a) the distance of the agent from its destination landmark, and (b) the percentage of times the agent accesses the privileged input (i.e., information from the other agents). Table 4 shows the relative distance as well as the percentage of times agents access information from other agents (in brackets).

**Results:** Table 4 compares an agent trained with proposed method to (Mordatch & Abbeel, 2017) and Infobot (Goyal et al., 2019). We also study how many times an agent access the privileged input. As shown in Table 4 (within brackets) the VBB can achieve better results, as compared to other methods, even when accessing the privileged input only less than 40% of the times.

## 7.5 INFORMATION CONTENT FOR VBB AND VIB

| Task | Infobot | Bernoulli- Reinforce | VBB |
|---|---|---|---|
| Navigation Env | 4.45 (100%) | 5.34 (74%) | 3.92 (**20%**) |
| Sequential MNIST | 3.56 (100%) | 3.63 (65%) | 3.22 (**46%**) |
| Model Based RL | 7.12 (100%) | 7.63 (65%) | 6.94 (**15%**) |

Table 5: The VBB performs better, as compared to the baselines. The VBB transmits a similar number of bits, while accessing privileged information a fraction of the time (in brackets % of times access to privileged information). Using REINFORCE to learn the parameter of the Bernoulli, does not perform as well as the proposed method.

**Channel Capacity:** We can quantify the average information transmission through both the VBB and the VIB in bits. The average information is similar to the conventional VIB, while the input

is accessed only a fraction of the time (the VIB accesses it 100% of the time). In order to show empirically that the VBB is minimizing information transmission (Eq. 1 in main paper), we measure average channel capacity $\mathbf{D}_{\text{KL}}(\mathbf{p}(\mathbf{z}|\mathbf{s}, \mathbf{g})\|\mathbf{r}(\mathbf{z}))$ numerically and compare the proposed method with the VIB, which must access the privileged input every time (See Table 5).

## 8  DISCUSSION

We demonstrated how the proposed variational bandwidth bottleneck (VBB) helps in generalization over the standard variational information bottleneck, in the case where the input is divided into a standard and privileged component. Unlike the VIB, the VBB does not actually access the privileged input before deciding how much information about it is needed. Our experiments show that the VBB improves generalization and can achieve similar or better performance while accessing the privileged input less often. Hence, the VBB provides a framework for adaptive computation in deep network models, and further study applying it to domains where reasoning about access to data and computation is an exciting direction for future work. Current limitation of the proposed method is that it assumes independence between standard input and the privileged input but we observe in practice assuming independence does not seem to hurt the results. Future work would be to investigate how we can remove this assumption.

## ACKNOWLEDGEMENTS

The authors acknowledge the important role played by their colleagues at Mila and RAIL throughout the duration of this work. AG is grateful to Alexander Neitz for the code of the environment used for model based experiments Fig. (2). AG is also grateful to Rakesh R Menon for pointing out the error and giving very useful feedback. The authors would also like to thank Alex Lamb, Nan Rosemary Ke, Olexa Bilaniuk, Jordan Hoffmann, Nasim Rahaman, Samarth Sinha, Shagun Sodhani, Devansh Arpit, Riashat Islam, Coline Devin, DJ Strousse, Jonathan Binas, Suriya Singh, Hugo Larochelle, Tom Bosc, Gautham Swaminathan, Salem Lahou for feedback on the draft. The authors are also grateful to the reviewers of ICML, NeurIPs and ICLR for their feedback. The authors are grateful to NSERC, CIFAR, Google, Samsung, Nuance, IBM, Canada Research Chairs, Canada Graduate Scholarship Program, Nvidia for funding, and Compute Canada for computing resources. We are very grateful to Google for giving Google Cloud credits used in this project.

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

## A    CONDITIONAL BOTTLENECK

In this section, we construct our objective function, such that minimizing this objective function minimizes $I(Y, G|S)$. Recall that the IB objective (Tishby et al., 2000) is formulated as the minimization of $I(Z, X) - \beta I(Z, Y)$, where $X$ refers to the input, $Y$ refers to the model output , $Z$ refers to compressed representation or the bottleneck. For the proposed method, we construct our objective as follows: we minimize the mutual information between privileged input and output given the standard input, $I(Y, G|S)$, to encode the idea that the we should avoid unnecessary access to privileged input $G$, and maximize the $I(Z, Y)$. Hence, for the VBB, using the data processing inequality (Cover & Thomas, 2006), this implies that

$$I(Z; G|S) \geq I(Y; G|S). \tag{5}$$

To obtain an upper bound on $I(G; Z|S)$, we must first obtain an upper bound on $I(G; Z|S = s)$, and then we average over $p(s)$. We get the following result:

$$I(G; Z|S = s) = \sum_{z,g} p(g|s) p(z|s, g) \log \frac{p(z|s, g)}{p(z|s)}, \tag{6}$$

We assume that the privileged input $G$ and the standard input $S$ are independent of each other, and hence $p(g|s) = p(g)$. we get the following upper bound:

$$
\begin{aligned}
I(G; Z|S = s) &\leq \sum_g p(g) \sum_z p(z|s, g) \log \frac{p(z|s, g)}{p_{prior}(z)} \\
&= \sum_g p(g) D_{\mathrm{KL}}(p(z|s, g) \| r(Z))
\end{aligned}
\tag{7}
$$

where the inequality in the last line is because we replace $p(z|s)$ with $p_{prior}(z)$. We also drop the dependence of the prior $z$ on the standard input $s$. While this loses some generality, recall that the predictive distribution $p(y|s, z)$ is already conditioned on $s$, so information about $s$ itself does not need to be transmitted through $z$ . Therefore, we have that $[p(Z|S) \| r(z)] \geq 0 \Rightarrow \sum_z p(z|s) \log p(z|s) \geq \sum_z p(z|s) \log r(z)$. Marginalizing over the standard input therefore gives us

$$
\begin{aligned}
I(Z; G|S) &\leq \sum_s p(s) \sum_g p(g) D_{\mathrm{KL}}[p(z|s, g) \| r(z)] \\
&= \sum_s p(s) \sum_g p(g) D_{\mathrm{KL}}[p(z|s, g) \| r(z)]
\end{aligned}
\tag{8}
$$

We approximate $p(z|s, g)$ as a weighted mixture of $p_{enc}(z_{enc}|s, g)$ and the normal prior such that $z \sim d_{cap} * (p_{enc}(z_{enc}|s, g)) + (1 - d_{cap}) * \mathcal{N}(0, 1)$. Hence, the weighted mixture $p(z|s, g)$ can be seen as a bound on the information bottleneck objective. Whenever we access the privileged input $G$, we pay an *information cost* (equal to $I(Z, G|S)$ which is bounded by $D_{\mathrm{KL}}(p(z|s, g) \| p_{prior}(z))$. Hence, the objective is to avoid accessing the privileged input, such that on average, the information cost of using the privileged input is minimal.

## B    TRACTABLE OPTIMIZATION OF KL OBJECTIVE

Here, we first show how the weighted mixture can be a bound on the information bottleneck objective.
Recall,

$$D_{\text{KL}}(P\|Q) = \sum_{x \in \mathcal{X}} P(x) \log \left( \frac{P(x)}{Q(x)} \right) \tag{9}$$

Hence, $D_{\text{KL}}(p(z|s, g)\|p_{prior}(z))$ where $p(z|s, g)$ is expressed as a mixture of direc delta and prior, and hence it can be written as

$$D_{\text{KL}}(p(z|s, g)\|r(z)) = D_{\text{KL}}(d_{cap} * p(z) + (1 - d_{cap}) * \delta(f(s, g))\|r(z)) \tag{10}$$

Further expanding the RHS using eq. 9, we get

$$D_{\text{KL}}(p(z|s, g)\|r(z)) = d_{cap} * \mathbb{E}_{p(z)}\big[\log p(z) - \log(d_{cap}p(z) + (1 - d_{cap}\underbrace{(\delta(f(s, g))}_{0})]$$
$$+ (1 - d_{cap}) \log p(f(s, g)) - \log(d_{cap}p(f(s, g)) + (1 - d_{cap})$$

Here, we can assume the $\delta(f(s, g))$ to be zero under the prior (as it is a Direc delta function). This can further be simplified to:

$$D_{\text{KL}}(p(z|s, g)\|r(z)) = d_{cap} * \mathbb{E}_{p(z)}\big[\log p(z) - \log(d_{cap}) - \log(p(z))\big]$$
$$+ (1 - d_{cap})[\log p(f(s, g)) - \log(d_{cap}p(f(s, g)) + (1 - d_{cap}))]$$

And hence, reducing the above term reduces t0 $D_{\text{KL}}(p(z|s, g)\|p_{prior}(z))$, our original objective.

## C    ANOTHER METHOD OF CALCULATING CHANNEL CAPACITY

In the main paper we show how can we evaluate channel capacity in a tractable way. The way we do is to learn a function $B(S)$ which determines channel capacity. Here's another way, which we (empirically) found that parameterizing the channel capacity network helps. In order to represent this function $B(S)$ which satisfies these constraints, we use an encoder of the form $(B(S) = p(z_{cap}|S))$ such that $z_{cap} \sim \mathcal{N}(z_{cap}|f^{\mu}(S), f^{\sigma}(S))$, where $S$ refers to the standard input, and $f^{\mu}, f^{\sigma}$ are learned functions (e.g., as a multi-layer perceptron) that outputs $\mu$ and $\sigma$ respectively for the distribution over $z_{cap}$. Here, $D_{KL}(B(S)|\mathcal{N}(0, 1))$ refers to the channel capacity of the bottleneck. In order to get a probability $prob$ out of $B(S)$, we convert $B(S)$ into a scalar $prob \in [0, 1]$ such that the $prob$ can be treated as a probability of accessing the privileged input.

$$prob = \text{Sigmoid}(\text{Normalization}(B(S))) \tag{11}$$

We perform this transformation by normalizing $B(S)$ such that $B(S) \in [-k, k]$, (in practice we perform this by clamping $B(S) \in [-2, 2]$) and then we pass the normalized $B(S)$ through a sigmoid activation function, and treating the output as a probability, $prob$, we access the privileged input with probability $prob$. Hence, the resulting distribution over z is a weighted mixture of accessing the privileged input $f_{enc}(s, g)$ with probability $prob$ and sampling from the prior with probability $1 - prob$. Here we assume prior to be $\mathcal{N}(0, 1)$, but it can also be learned. At test time, using $prob$, we can sample from the Bernouilli distribution $b \sim Bernoulli(prob)$ to decide whether to access the privileged input or not.

## D    MULTIAGENT COMMUNICATION

**Experimental Setup:** We use the setup proposed by Mordatch & Abbeel (2017). The environment consists of N agents and M landmarks. Both the agents and landmarks exhibit different characteristics such as different color and shape type. Different agents can act to move in the environment. They can also be affected by the interactions with other agents. Asides from taking physical actions, agents

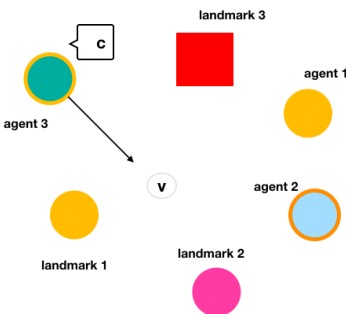

Figure 4: Multiagent Communciation: The environment consists of N agents and M landmarks. Both the agents and landmarks exhibit different characteristics such as different color and shape type. Different agents can act to move in the environment. They can also act be effected by the interactions with other agents. Asides from taking physical actions, agents communicate with other agents using verbal communication symbols $c$.

communicate with other agents using verbal communication symbols. Each agent has a private goal which is not observed by another agent, and the goal of the agent is grounded in the real physical environment, which might include moving to a particular location, and could also involve other agents (like requiring a particular agent to move somewhere) and hence communication between agents is required.

Each agent performs actions and communicates utterances according to a policy, which is identically instantiated for all of the agents in the environment, and also receive the same reward signal. This policy determines both the actions and communication protocols. We assume all agents have identical action and observation spaces and receive the same reward signal. We consider the cooperative setting, in which the problem is to find a policy that maximizes expected return for all the agents.

## E    SPATIAL REASONING

In order to study generalization across a wide variety of environmental conditions and linguistic inputs, (Janner et al., 2018) develop an extension of the puddle world reinforcement learning benchmark. States in a 10 X 10 grid are first filled with either grass or water cells, such that the grass forms one connected component. We then populate the grass region with six unique objects which appear only once per map (triangle, star, diamond, circle, heart, and spade) and four non-unique objects (rock, tree, horse, and house) which can appear any number of times on a given map. We followed the same experimental setup and hyperparameters as in (Janner et al., 2018).

Here, an agent is rewarded for reaching the location specified by the language instruction. Agent is allowed to take actions in the world. Here the goal is to be able to generalize the learned representation for a given instruction such that even if the environment observations are rearranged, this representation is still useful. Hence, we want to learn such representations that ties observations from the environment and the language expressions. Here we consider the Puddle World Navigation map as introduced in (Janner et al., 2018). We followed the same experiment setup as (Janner et al., 2018). Here, the current state of the agent acts as a standard input. Based on this, agent decides to access the privileged input.

We start by converting the instruction text into a real valued vector using an LSTM. It first convolves the map layout to a low-dimensional repesentation (as opposed to the MLP of the UVFA) and concatenates this to the LSTM's instruction embedding (as opposed to a dot product). These concatenated representations are then input to a two layered MLP. Generalization over both environment configurations and text instructions requires a model that meets two desiderata. First, it must have a flexible representation of goals, one which can encode both the local structure and global spatial attributes inherent to natural language instructions. Second, it must be compositional, in order to learn a generalizable representation of the language even though each unique instruction will only be observed with a single map during training. Namely, the learned representation for a given instruction should still be useful even if the objects on a map are rearranged or the layout is changed entirely.

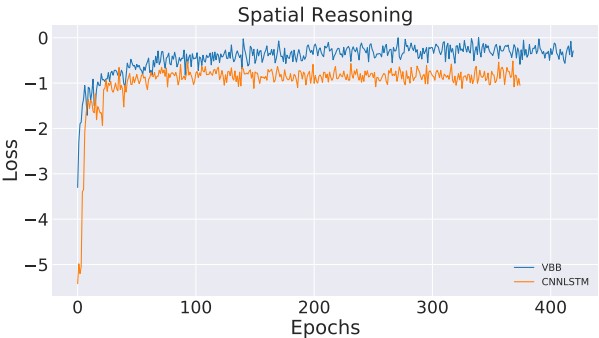

Figure 5: Spatial Reasoning

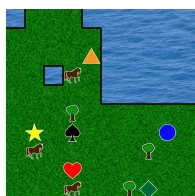

Figure 6: Puddle world navigation

# F    RECURRENT VISUAL ATTENTION - LEARNING BETTER FEATURES

The goal of this experiment is to study if using the proposed method enables learning a dynamic representation of an image which can be then used to *accurately* classify an image. In order to show this, we follow the setup of the Recurrent Attention Model (RAM) (Mnih et al., 2014). Here, the attention process is modeled as a sequential decision process of a goal-directed agent interacting with the visual image. A recurrent neural network is trained to process the input sequentially, attending to different parts within the image one at a time and hence combining information from these different parts to build up a dynamic representation of the image. The agent incrementally combines information because of attending to different parts and then chooses this integrated information to choose where next to attend to. In this case, the information due to attending at a particular part of the image acts as a standard input, and the information which is being integrated over time acts as a privileged input, which is then used to select where the model should attend next. The entire process repeats for N steps (for our experiment N = 6). FC denotes a fully connected network with two layers of rectifier units, each containing 256 hidden units.

| Model | MNIST | 60 * 60 Cluttered MNIST |
|---|---|---|
| FC (2 layers) | 1.69% | 11.63% |
| RAM Model (6 locs) | 1.55% | 4.3% |
| VIB (6 locs) | 1.58% | 4.2% |
| VBB (6 locs) (Ours) | 1.42% | 3.8% |

Table 6: Classification error results (Mnih et al., 2014). Averaged over 3 random seeds.

**Quantitative Results:**  Table 6 shows the classification error for the proposed model, as well as the baseline model, which is the standard RAM model. For both the proposed model, as well as the RAM model, we fix the number of locations to attend to equal to 6. The proposed method outperforms the standard RAM model.

## G  ALGORITHM IMPLEMENTATION DETAILS

We evaluate the proposed framework using Advantage Actor-Critic (A2C) to learn a policy $\pi_\theta(a|s, g)$ conditioned on the goal. To evaluate the performance of proposed method, we use a range of maze multi-room tasks from the gym-minigrid framework (Chevalier-Boisvert & Willems, 2018) and the A2C implementation from (Chevalier-Boisvert & Willems, 2018). For the maze tasks, we used agent's relative distance to the absolute goal position as "goal".

For the maze environments, we use A2C with 48 parallel workers. Our actor network and critic networks consist of two and three fully connected layers respectively, each of which have 128 hidden units. The encoder network is also parameterized as a neural network, which consists of 1 fully connected layer. We use RMSProp with an initial learning rate of 0.0007 to train the models, for both InfoBot and the baseline for a fair comparison. Due to the partially observable nature of the environment, we further use a LSTM to encode the state and summarize the past observations.

## H  MINIGRID ENVIRONMENTS FOR OPENAI GYM

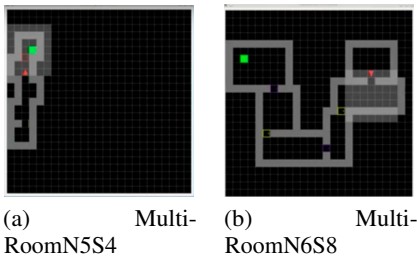

(a)              Multi-
RoomN5S4

(b)              Multi-
RoomN6S8

Figure 7: Partially Observable MultiRoomsNXSY environments

The MultiRoom environments used for this research are part of MiniGrid, which is an open source gridworld package. This package includes a family of reinforcement learning environments compatible with the OpenAI Gym framework. Many of these environments are parameterizable so that the difficulty of tasks can be adjusted (e.g., the size of rooms is often adjustable).

### H.1  THE WORLD

In MiniGrid, the world is a grid of size NxN. Each tile in the grid contains exactly zero or one object. The possible object types are wall, door, key, ball, box and goal. Each object has an associated discrete color, which can be one of red, green, blue, purple, yellow and grey. By default, walls are always grey and goal squares are always green.

### H.2  REWARD FUNCTION

Rewards are sparse for all MiniGrid environments. In the MultiRoom environment, episodes are terminated with a positive reward when the agent reaches the green goal square. Otherwise, episodes are terminated with zero reward when a time step limit is reached. In the FindObj environment, the agent receives a positive reward if it reaches the object to be found, otherwise zero reward if the time step limit is reached.

The formula for calculating positive sparse rewards is $1 - 0.9 * (step\_count/max\_steps)$. That is, rewards are always between zero and one, and the quicker the agent can successfully complete an episode, the closer to 1 the reward will be. The $max\_steps$ parameter is different for each environment, and varies depending on the size of each environment, with larger environments having a higher time step limit.

---

https://github.com/maximecb/gym-minigrid

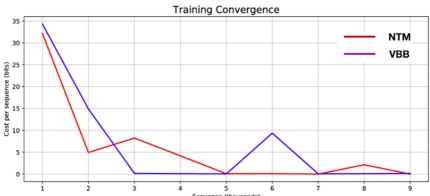

Figure 8: Copying Task

## H.3 ACTION SPACE

There are seven actions in MiniGrid: turn left, turn right, move forward, pick up an object, drop an object, toggle and done. For the purpose of this paper, the pick up, drop and done actions are irrelevant. The agent can use the turn left and turn right action to rotate and face one of 4 possible directions (north, south, east, west). The move forward action makes the agent move from its current tile onto the tile in the direction it is currently facing, provided there is nothing on that tile, or that the tile contains an open door. The agent can open doors if they are right in front of it by using the toggle action.

## H.4 OBSERVATION SPACE

Observations in MiniGrid are partial and egocentric. By default, the agent sees a square of 7x7 tiles in the direction it is facing. These include the tile the agent is standing on. The agent cannot see through walls or closed doors. The observations are provided as a tensor of shape 7x7x3. However, note that these are not RGB images. Each tile is encoded using 3 integer values: one describing the type of object contained in the cell, one describing its color, and a flag indicating whether doors are open or closed. This compact encoding was chosen for space efficiency and to enable faster training. The fully observable RGB image view of the environments shown in this paper is provided for human viewing.

## H.5 LEVEL GENERATION

The level generation in this task works as follows: (1) Generate the layout of the map (X number of rooms with different sizes (at most size Y) and green goal) (2) Add the agent to the map at a random location in the first room. (3) Add the goal at a random location in the last room. **MultiRoomN$X$S$Y$** - In this task, the agent gets an egocentric view of its surroundings, consisting of $3\times3$ pixels. A neural network parameterized as MLP is used to process the visual observation.

## I MEMORY ACCESS - DECIDING WHEN TO ACCESS MEMORY

Here, the privileged input involves accessing information from the external memory like neural turing machines (NTM) (Sukhbaatar et al., 2015; Graves et al., 2014). Reading from external memory is usually an expensive operation, and hence we would like to minimize access to the external memory. For our experiments, we consider external memory in the form of neural turning machines. NTM processes inputs in sequences, much like a normal LSTM but NTM can allow the network to learn by accessing information from the external memory. In this context, the state of controller (the NTM's controller which processes the input) becomes the standard input, and based on this (the standard input), we decide the channel capacity, and based on channel capacity we decide whether to read from external memory or not. In order to test this, we evaluate our approach on copying task. This task tests whether NTMs can store and recall information from the past. We use the same problem setup as (Graves et al., 2014). As shown in fig 8, we found that we can perform slightly better as compared to NTMs while accessing external memory only **32%** of the times.

## J    HYPERPARAMETERS

The only hyperparameter we introduce with the variational information bottleneck is $\beta$. For both the VIB baseline and the proposed method, we evaluated with 5 values of $\beta$: 0.01, 0.09, 0.001, 0.005, 0.009.

### J.1    COMMON PARAMETERS

We use the following parameters for lower level policies throughout the experiments. Each training iteration consists of 5 environments time steps, and all the networks (value functions, policy , and observation embedding network) are trained at every time step. Every training batch has a size of 64. The value function networks and the embedding network are all neural networks comprised of two hidden layers, with 128 ReLU units at each hidden layer.

All the network parameters are updated using Adam optimizer with learning rate $3 \cdot 10^{-4}$.

Table 7 lists the common parameters used.

| Parameter | Value |
|---|---|
| learning rate | $3 \cdot 10^{-4}$ |
| batch size | 64 |
| discount | 0.99 |
| entropy coefficient | $10^{-2}$ |
| hidden layers (Q, V, embedding) | 2 |
| hidden units per layer (Q, V, embedding) | 128 |
| Bottleneck Size | 64 |
| RNN Hidden Size | 128 |
| $\beta$ | 0.001/0.009/0.01/0.09 |

Table 7: Shared parameters for benchmark tasks

## K    ARCHITECTURAL DETAILS

For our work, we made sure to keep the architecture detail as similar to the baseline as possible.

- **Goal Driven Navigation:** Our code is based on open source gridworld package `https://github.com/maximecb/gym-minigrid`.
- **Multiagent Communication:** Our code is based on the following open source implementation. `https://github.com/bkgoksel/emergent-language`.
- **Access to External Memory:** Our code is based on the following open source implementation of Neural Turing Machines. `https://github.com/loudinthecloud/pytorch-ntm`
- **Spatial Navigation:** Our code is based on the following open source implementation of `https://github.com/JannerM/spatial-reasoning`.

The only extra parameters which our model is introduce is related to the channel capacity network, which is parameterized as a neural network consisting of 2 layers of 128 dimensions each (with ReLU non-linearity).

