# OpenReview forum: "The Variational Bandwidth Bottleneck: Stochastic Evaluation on an Information Budget"
_ICLR.cc/2020/Conference — Accept (Poster)_

### Official Review · AnonReviewer2 · 2019-10-23
**Official Blind Review #2**

**Rating:** 6

**Review:**

Summary - The paper proposes an approach, called the Variational Bandwidth Bottleneck (VBB), capable of compressing only a part of the input and still learn representations that are informative of the output. The approach is motivated from the following perspective -- there might be situations where a ``part’’ of the input is privileged in the sense that it may be costly / wasteful to maintain access to all the time, thereby rendering the standard IB pipeline infeasible (as it requires unrestricted access to the entire input). By breaking down the input into standard (always available) and privileged (not always available) components, the paper proposes a module that decides whether to access the privileged input during compression based on the standard input. The goal is to be able to decide when to access privileged information and not how to break the overall input into standard and privileged components. The approach tackles a narrow subset of problems compared to the standard information bottleneck. The authors show the applicability of the proposed approach in reinforcement learning setups -- specifically, (1) when to access an expensive model-based planner for goal-driven navigation; (2) when to access goal-information in goal-driven navigation and (3) treating communication in a multi-agent cooperative setting as ``privileged’’ information. The experimental results demonstrate that VBB accesses privileged information in a feasible and minimal manner and results in better generalization performance.

Strengths

- Apart from the flaws (highlighted in weaknesses), the paper is well-written and generally easy to follow.

- The problem statement is well-motivated. The authors did a good job of first identifying when the standard IB approach would be infeasible / costly -- even though representations are compressed and relevant, computing them still requires unrestricted access to input -- and then motivating the need for selective access to information while decision-making. The problem is well-grounded in the experimental settings the authors provide results for.

- Apart from the concerns (highlighted in weaknesses), the experimental results generally support the claims of the paper. Results in Sec. 7.1 (loosely) demonstrate that VBB accesses privileged information states where degree of freedom in terms of possible trajectories is higher -- this result, although not explored in it’s entirety, also correlates with notions of decision-states (as pointed out by authors) and bottleneck states in the existing literature. Results in Sec. 7.2 demonstrate improved generalization performance in terms of transfer. Additionally, results in Table.3 suggest that VBB accesses privileged information the least number of times. Similarly, results in Sec. 7.3 indicate VBB results in improvements in performance in the multi-agent setting while resulting in minimal communication among the agents. The baselines being compared with in the paper are also reasonable.

- The problem statement and the proposed approach have some degree of novelty. Most works along the lines of restricting access to relevant information still assume unrestricted access to the `entire’ information.

Weaknesses

That being said, the paper does have some flaws / clarity issues that make several associated details confusing when judged in context of prior work. These concerns (in addition to the strengths highlighted above) mostly surrounding experiments form the basis of my rating and addressing these would definitely make the paper stronger.

- One of the underlying motivations / intuitions behind restricting access to the privileged information is that fact that -- “..avoid accessing the privileged input because we want to generalize with respect to it..”. This statement in isolation is misleading and makes things unclear. From a complete generalization standpoint, preventing the encoder from overfitting to the input (‘privileged’ or ‘standard’) can be addressed by just tracking the relevance (predictive performance) based on the representations on a held-out set. Unless I am missing something, generalization of the learned representations in standard IB becomes major problem only when we restrict (costly or otherwise) access to the privileged input. The statement seems to suggest that learning representations completely and generally agnostic to the privileged input is a good idea. However, this is only true when at test-time, the privileged input may be significantly different than what was seen during training time -- a different model-based planner / entirely different goal-specifications, etc. Could the authors comment more on this and reframe the intuition wherever applicable?

- Last 3 lines of the 1st paragraph in section 3 (page 2) are incomplete -- “...constraining the channel capacity .. is permitted to differ from the prior r(Z)....”. The difference in the posterior p(Z|X) and the prior r(Z) quantify the channel capacity only under expectation over the distribution over inputs -- p(X). Therefore, in practice, it is also governed by the empirical distribution of the data (see page 4 of https://arxiv.org/pdf/1612.00410.pdf). The authors should change the lines to reflect the same.

- One of the key contributions of the paper is to develop a mechanism where one isn’t required to access the privileged input all the time -- depending on the standard input, one can decide whether to access the privileged information. This manifests in form of a mixture-distribution over a dirac-delta transformation of the deterministic encoder and a prior distribution. At inference-time, the channel capacity (d_cap, based on the standard input) can be used to decide whether to access the privileged information or not. There’s lack of clarity in terms of what happens at training time -- is it the case that (1) d_cap is computed, f(s, g) is also computed and based on the KL-term in Eq (3), B(S) is incentivized to generate d_cap that results in less frequent access the privileged information (d_cap < 0.5 for the most part) or (2) d_cap is computed, a sample is drawn from the bernoulli and z is sampled from either r(Z) or \del_(f(S, G) and KL is either 0 (if z ~ r(Z)) or a finite value (if z ~  \del_(f(S, G))? (1) involves an exact computation and always requires access to G (privileged input) during training whereas (2) is approximate and does not always require access to G. It’s unclear what the pipeline is from the paper. Can the authors clarify this? Are there specific reasons why (1) / (2) was chosen?

- Experimental Results: Highlighting these below:
     - In the experimental setting of Figure 2, do the authors notice any (1) qualitative differences in terms of where the agent accesses the output of the planner when InfoBot is used (adapted to this setting) and (2) quantitative differences (Table. 2) in terms of how often does InfoBot and the other proposed baselines access the planner output at junctions and hallways? Junctions in 2D mazes can potentially be identified as important decision states using simpler approaches. As a demonstrative experiment, the key point to be made using this experiment seems to be how often does VBB access the planner output compared to InfoBot and other baselines.

    - In Sec. 7.2, it is unclear how VBB is being used to generalize to novel environments -- as in, is the pipeline same as InfoBot, where a frozen encoder is used to provide an exploration incentive?

    - While the generalization results in Table. 2 are impressive in terms of success values, I think it lacks a few numbers (assuming the transfer pipeline to novel environments is same as InfoBot) -- given that the environments being tested on are different from InfoBot, how well do count-based exploration and goal-conditioned A2C baselines perform? This is to understand whether InfoBot is the right thing to compare with in these environments.

    - Furthermore, for the goal-driven navigation set of experiments can the authors report comparisons in terms of sample-efficiency as well -- how do success-rates and average task-returns vary with time-steps of training?

    - For results in Sec. 7.3, Table. 4, I would encourage the authors to compare with IC3Net (https://arxiv.org/pdf/1812.09755.pdf) which learns “when to communicate” in a multi-agent setting irrespective of whether it’s a cooperative situation or not.

Reasons for rating

Apart from the points mentioned above, I don’t have major weaknesses to point out. The paper is generally easy to follow and the proposed approach and problem statement is well-grounded and somewhat novel. However, the paper suffers from lack of experimental details and comparisons (and other weaknesses highlighted above) and therefore I am inclined towards my current rating. Addressing those would significantly benefit the paper and help me in  increasing my score.


**Experience Assessment:**

I have read many papers in this area.

**Review Assessment: Checking Correctness Of Derivations And Theory:**

I assessed the sensibility of the derivations and theory.

**Review Assessment: Checking Correctness Of Experiments:**

I assessed the sensibility of the experiments.

**Review Assessment: Thoroughness In Paper Reading:**

I read the paper thoroughly.

---

> ### Author Response · Authors · 2019-11-07
> **Response to Reviewer (1/3)**
>
>
> We thank the reviewer for their feedback and their generally positive assessment of our work. We believe this would further improve the readability of our work. Thanks!!
>
>
> We have ran additional experiments requested by the reviewer. To summarize.
> (a) We ran the comparison to I3C as requested by reviewer.
> (b) We have also compared to count based exploration as requested by reviewer.
>
> “..avoid accessing the privileged input because we want to generalize with respect to it..”
>
> The VBB is indeed meant to provide robustness to distributional shift. While this can happen, as the reviewer says, in cases where there "a different model-based planner / entirely different goal-specifications," this kind of distributional shift more commonly occurs when we study generalization to related but distinct problem instances. For example, in the settings considered in our experiments, we train on procedurally generated mazes with different sizes (like different number of rooms, and size of different rooms) during training and testing.
>
> “the authors should change the lines to reflect the same.”
>
> Thanks for pointing this out. Its indeed an omission from our side.
>
> “It’s unclear what the pipeline is from the paper. Can the authors clarify this? ”
>
> We follow the first pipeline.
> (1) d_cap is computed.
> (2)  f(s, g) is also computed and based on the KL-term in Eq (3).
> (3) B(S) is incentivized to generate d_cap that results in less frequent access the privileged information.
> (4) During test time, we sample from the bernouilli treating d_prob as the probability. Main motivation was, it resonates well with our result in Proposition 1, and during training everything is still differentiable ( though we do need to access privileged information during training)

---

> > ### Author Response · Authors · 2019-11-07
> > **VBB and InfoBot (2/3)**
> >
> > “as in, is the pipeline same as InfoBot, where a frozen encoder is used to provide an exploration incentive? it is unclear how VBB is being used to generalize to novel environments”
> >
> > Authors in InfoBot evaluate generalization by 2 ways. (1) using policy transfer, and (2) Using the encoder for providing exploration bonus in a new environment.  In the proposed method, we evaluate generalization by directly transferring policy in a new environment (and not by freezing the encoder, and providing exploration bonus). Our motivation is we want to test whether agent can generalize better by dynamically deciding when to access the privileged input.  Hence, we  demonstrate that training an agent with a VBB leads to more effective policy transfer (as compared to just having a goal conditioned bottleneck). We train policies on smaller versions of the MiniGrid environments, but evaluate them on larger versions throughout training. This is similar to section 4.2 in InfoBot paper. (https://openreview.net/pdf?id=rJg8yhAqKm).
> >
> > “Comparison to Count based exploration”
> >
> > We compare the proposed method to Goal Conditioned A2C (UVFA) as in Table 2, as well as Count based exploration as requested by the reviewer. For this paper, we consider direct policy transfer where we train policy in one environment and test it in other environments.
> >
> > Method                                   FindObjS7       FindObjS10
> > VBB                                           73% ± 2%          57% ± 4%
> > Goal conditioned A2C            40% ± 2%          24% ± 3%
> > Count Based Exploration      45% ± 3%          38% ± 5%

---

> > > ### Author Response · Authors · 2019-11-07
> > > **Comparison with IC3net (3/3)**
> > >
> > > “the authors to compare with IC3Net”
> > >
> > > We thank the reviewer for the reference. As asked by reviewer, we ran the IC3Net baseline for our task by varying different number of agents. IC3Net performs better for 6 agents, but VBB scales better (and outperforms IC3Net baseline) when there are more number of agents (like 10 or 15). We also study how many times an agent access the privileged input (in brackets). For all the scenarios, (6 agents, 10 agents and 15 agents), proposed method (VBB) access privileged input less number of times as compared to IC3Net. We think the reason VBB scales better (in this scenario is because VBB is fully differentiable, but in IC3Net requires a non-differentiable decision. Future work would be how to improve IC3Net using ideas from VBB. (Less is better).
> > >
> > > Method              6 agents                              10 agents                          15 agents
> > > VBB (ours)        4.72 (23%)                            5.22 (34%)                            5.76 (38%)
> > > IC3net               4.68 (34%)                            5.28 (42%)                           5.91 (45%)
> > >
> > > Thanks again for your time in reviewing our submission. We really appreciate it.

---

> ### Author Response · Authors · 2019-11-13
> **Updated Impression ?**
>
> R2, We believe, we have addressed your concerns and clarified some of your points.
>
> We hope to have changed your assessment of our work for the better; should that not be the case, please do not hesitate to get in touch with us.
>
> Thanks for your time.

---

> ### Author Response · Authors · 2019-11-15
> **Feedback ?**
>
> Dear reviewer #2,
>
> We’d like to thank you again for your review and feedback! We have provided response to your questions. In particular, we clarified connections to InfoBot, did comparisons to Count Based exploration, some more experiments with I3C paper.
>
> Would you have any other questions regarding the rebuttal?
>
> Since the review discussion period is going to end, we would appreciate any feedback that you might have. We would be very happy to provide more clarifications, if any.
>
> Many thanks again for your review, feedback and time. We appreciate it.

---

> ### Comment · AnonReviewer2 · 2019-11-15
> **Thanks for responding to the comments!**
>
> Thanks to the authors for providing detailed comments and justifications wherever applicable and apologies for the late reply. I'll discuss / respond to the comments of the authors below.
>
> > The VBB is indeed meant to provide robustness to distributional shift. While this can happen, as the reviewer says, in cases where there "a different model-based planner / entirely different goal-specifications," this kind of distributional shift more commonly occurs when we study generalization to related but distinct problem instances. For example, in the settings considered in our experiments, we train on procedurally generated mazes with different sizes (like different number of rooms, and size of different rooms) during training and testing.
>
> Thanks for responding to this. Just to clarify, my point here was that the degree to which a model / algorithm should be agnostic to the privileged input depends on the problem setting (as rightly pointed out by the authors). My intention behind the comment was to encourage the authors to soften the statement / claims around the robustness aspect of the motivation since beyond "the cost of accessing privileged input", robustness w.r.t. privileged input doesn't seem to be the problem in the experimental settings presented in the paper.
>
> Thanks for clarifying the training pipeline and making it clear that access to privileged information is still always required during training (but not during testing). I think out of the two options that I discussed in the comment, (1) seems the more reasonable thing to try first. My comment was motivated by trying to draw parallels with sampling procedures in mixture models.
>
> Thanks for making the transfer pipeline clear.
>
> Thanks for presenting the comparisons to the count-based and goal-conditioned A2C baseline in terms of success values.
>
> Thanks for including the comparison with IC3Net. Comparisons across a range over the number of agents is especially interesting. The fact that VBB scales better relative to IC3Net with number of agents does indeed demonstrate the utility of VBB.
>
> The response provided by the authors cover most of my points, except for one -- comparisons w.r.t. sample-efficiency of VBB versus InfoBot (and other approaches). While success and frequency of accessing privileged information are interesting metrics to look at, comparisons w.r.t. sample-efficiency would reflect if restricted access to privileged information via VBB results in faster training -- by very carefully accessing privileged information only at specific states.
>
> Given the responses to my comments, I am generally supportive of the paper.

---

> > ### Author Response · Authors · 2019-11-15
> > **comparisons w.r.t. sample-efficiency of VBB versus InfoBot + revise score ?**
> >
> > We appreciate the response by the reviewer.
> >
> > "comparisons w.r.t. sample-efficiency of VBB versus InfoBot (and other approaches). While success and frequency of accessing privileged information are interesting metrics to look at, comparisons w.r.t. sample-efficiency would reflect if restricted access to privileged information via VBB results in faster training -- by very carefully accessing privileged information only at specific states."
> >
> > We did not observe any change in sample efficiency b/w InfoBot as well as the proposed method.
> > We note that in the current work goal of the proposed method is not to obtain better sample efficiency, but to generalize better with respect to the privileged input.  Future work would investigate how to combine efficiently the proposed information regularizer with other information theoretic exploration methods which can also improve sample efficiency.
> >
> > Another interesting thing which we observed was, that in our navigation experiments we measured percentage of time steps on which (Infobot as well as the proposed method) access the goal information when the agent is near the junction point (or branching points) in the maze. We observe that the
> > proposed method learns to access the privileged input (in this case, the goal) only when necessary, as compared to Infobot baseline. This even holds when we evaluate on different task distribution as compared to what it was trained on. (For more details, ref. Tab 3 in the main paper). This further shows, that the encoder in "Infobot" can be improved by conditionally computing whether the privileged information should be accessed or not (as in the proposed method).
> >
> >
> > "Given the responses to my comments, I am generally supportive of the paper."
> >
> > We appreciate that the reviewer supports the paper. Would reviewer like to revise their score ?
> >
> > Thanks a lot for your time. :)

---

> > > ### Comment · AnonReviewer2 · 2019-11-15
> > > **Thanks again for responding to the comments!**
> > >
> > > >We did not observe any change in sample efficiency b/w InfoBot as well as the proposed method.
> > > We note that in the current work goal of the proposed method is not to obtain better sample efficiency, but to generalize better with respect to the privileged input.
> > >
> > > Thanks for responding to this. The observation (assuming "any change in sample efficiency b/w InfoBot" indicates no significant difference) reported in the first sentence highlights that VBB isn't worse compared to InfoBot in terms of sample-efficiency. This, supported by improvements in success-rate seem to indicate a somewhat positive result -- VBB takes as much time to attain better performance relative to InfoBot but accesses privileged information sparsely. Regardless, while it's understandable that the goal of the proposed approach may not have been centered towards sample efficiency, I would like to point out that one seemingly reasonable expectation is that learning when to access privileged information in a minimal manner would lead to increased sample efficiency -- learning when and where to access privileged information would avoid wasteful computation and therefore, may impact sample-efficiency.
> > >
> > > In light of the responses, I have updated my score to reflect the increased rating.

---

### Official Review · AnonReviewer1 · 2019-10-28
**Official Blind Review #1**

**Rating:** 6

**Review:**

This paper proposes a type of conditional Information Bottleneck (IB) that addresses the following problem: given that some features may be expensive to obtain for use in prediction, when should they be obtained such that the overall benefit outweighs the cost? A variant of the IB is proposed to model this question. However, optimization is intractable. The paper replaces a certain non-differentiable operation by a deterministic neural network which outputs the probability of seeking the expensive features. The main application here is reinforcement learning, where an agent could compute some plan or communicate with other agents at a cost, and the goal is to solve the task more efficiently while making use of this additional information. It is shown that the proposed method, VBB, makes judicious use of the limited number of costly feature acquisitions it makes, resulting in improved task performance across 3 tasks.

I am not an expert on the topic, but I find that this paper is well-written and tackles a basic question with effective methods that work well in practice.

However, since the problem is so basic, I wonder how it connects to active learning or more specifically "active feature acquisition":

-Saar-Tsechansky, Maytal, Prem Melville, and Foster Provost. "Active feature-value acquisition." Management Science 55.4 (2009): 664-684.
-Shim, Hajin, Sung Ju Hwang, and Eunho Yang. "Joint active feature acquisition and classification with variable-size set encoding." Advances in Neural Information Processing Systems. 2018.
-Ma, Chao, et al. "Eddi: Efficient dynamic discovery of high-value information with partial VAE." arXiv preprint arXiv:1809.11142 (2018).

Minor:
- "((Bahdanau et al., 2014; Mnih et al., 2014; Xu et al., 2015))": double parentheses
- "minimizing unnecessary access? . We compare": drop the . after ?
- "to dynamically adjusts" --> "to dynamically adjust"
- "The agent always access the" --> "The agent always accesses the"
- "Tables 3a, 3b compares": no such tables in the paper
- "each method acsess the" --> "each method accesses the"

**Experience Assessment:**

I do not know much about this area.

**Review Assessment: Checking Correctness Of Derivations And Theory:**

I did not assess the derivations or theory.

**Review Assessment: Checking Correctness Of Experiments:**

I assessed the sensibility of the experiments.

**Review Assessment: Thoroughness In Paper Reading:**

I read the paper at least twice and used my best judgement in assessing the paper.

---

> ### Author Response · Authors · 2019-11-07
> **Response to Reviewer**
>
> We thank the reviewer for their feedback and their generally positive assessment of our work.
>
> “I wonder how it connects to active learning or more specifically "active feature acquisition"
>
> We thank the reviewer for pointing this out. We agree that their are some intriguing connections to “active feature acquisition”. More generally, the  efficient use of limited computational resources is probably an important ingredient.  There’s a whole field of rational meta-reasoning which decide which computations to perform but this is computationally intractable. VBB provides a tractable way to maintain a trade-off b/w “cost” of performing a computation, and “value” of computation, in the sense perform a computation if value of computation exceeds that of the cost of computation. Future work would be to rigorously study this in active feature acquisition scenario.
>
>
> “Typos”
>
> We thank the reviewer for pointing out these corrections. We would update it in the next version of the paper.
>
> “References”
>
> We thank the reviewer for those references. We’ll look into them.
>
> Are there any other experiments or clarifications that we can provide such that would help to improve the understanding of our work and hence making it more likely for you to raise your score?
>
> Thanks again for your time in reviewing our submission. We really appreciate it.

---

> > ### Comment · AnonReviewer1 · 2019-11-13
> > **Rebuttal feedback**
> >
> > Thank you for the response. Given my limited expertise on the topic, I am unlikely to increase my Rating. However, I am generally supportive of the paper.

---

### Public Comment · ~Rakesh_R_Menon2 · 2020-01-21
**Clarification on derivation**

Nice work!

However, while going through the derivation of equation 3 (in appendix section "EASY OPTIMIZATION OF KL OBJECTIVE", not "Tractable Optimization of the KL Objective" as mentioned in the main paper [typo]), I found that a few parantheses were missing.

By deriving the expression by hand, I got a different expression (with the same terms though). I was wondering if the author(s) could provide some clarification.

Equation (3) from paper (calling b = p(f(s,g))):

- d_cap * E_z[ log(d_cap) ]  + (1-d_cap) * [ log(b) ]  - log( d_cap*b ) + (1-d_cap)

However, what I got was (applying eq 9 to eq 10):

d_cap * E_z[ log(d_cap) ]  - (1-d_cap) * [ log(b) - log( d_cap*b + (1-d_cap) ) ]
                                                                                     -------------------a--------------
                                                                    ----------------------b--------------------------
----------------------------------------------c------------------------------------------------------


In particular note:
(a) the last two terms come under the same logarithm.
(b) the three terms combined are multiplied by (1-d_cap).
(c) The sign is flipped.


Thanks in advance!

---

> ### Author Response · Authors · 2020-01-21
> **Thanks.**
>
> Hello Rakesh,
>
> Thanks for taking time and reading our work.
>
> you are right regarding missing brackets. Regarding the sign, there's \beta in the objective,
> depending upon which it can either be positive or negative.

---

### Decision · Program_Chairs · 2019-12-19

**Decision:**

Accept (Poster)

**Comment:**

Existing implementation of information bottleneck need access to privileged information which goes against the idea of compression. The authors propose variational bandwidth bottleneck which estimates the value of the privileged information and then stochastically decided whether to access this information or not. They provide a suitable approximation and show that their  method improves generalisation in RL while reducing access to expensive information.

These paper received only two reviews. However, both the reviews were favourable. During discussions with the AC the reviewers acknowledged that most of their concerns were addressed. R2 is still concerned that VBB does not result in improvement in terms of sample efficiency. I request the authors to adequately address this in the final version. Having said that, the paper does make other interesting contributions, hence I recommend that this paper should be accepted.